# Natural Compounds Purified from the Leaves of *Aristotelia chilensis*: Makomakinol, a New Alkaloid and the Effect of Aristoteline and Hobartine on Na_V_ Channels

**DOI:** 10.3390/ijms242115504

**Published:** 2023-10-24

**Authors:** Rebeca Pérez, Claudia Figueredo, Viviana Burgos, Jaime R. Cabrera-Pardo, Bernd Schmidt, Matthias Heydenreich, Andreas Koch, Jennifer R. Deuis, Irina Vetter, Cristian Paz

**Affiliations:** 1Laboratory of Natural Products & Drug Discovery, Center CEBIM, Universidad de La Frontera, Av. Francisco Salazar 01145, Temuco 4780000, Chile; perezcolladorebeca@gmail.com (R.P.); yanclaudia17@gmail.com (C.F.); 2Departamento de Ciencias Biológicas y Químicas, Facultad de Recursos Naturales, Universidad Católica de Temuco, Rudecindo Ortega 02950, Temuco 4780000, Chile; vburgos@uct.cl; 3Laboratorio de Química Aplicada y Sustentable (LabQAS), Departamento de Química, Facultad de Ciencias, Universidad del Bío-Bío, Concepción 4081112, Chile; jacabrera@ubiobio.cl; 4Institut für Chemie, Universität Potsdam, Karl-Liebknecht-Str. 24-25, D-14476 Potsdam, Germany; bernd.schmidt@uni-potsdam.de (B.S.); kochi@uni-potsdam.de (A.K.); 5Institute for Molecular Bioscience, The University of Queensland, St Lucia, QLD 4072, Australia; j.deuis@uq.edu.au (J.R.D.); i.vetter@imb.uq.edu.au (I.V.); 6School of Pharmacy, The University of Queensland, Woolloongabba, QLD 4102, Australia

**Keywords:** indole alkaloids, makomakinol, aristoteline, hobartine, *Aristotelia chilensis*, Na_V_1.8

## Abstract

*Aristotelia chilensis* or “maqui” is a tree native to Chile used in the folk medicine of the Mapuche people as an anti-inflammatory agent for the treatment of digestive ailments, fever, and skin lesions. Maqui fruits are black berries which are considered a “superfruit” with notable potential health benefits, promoted to be an antioxidant, cardioprotective, and anti-inflammatory. Maqui leaves contain non-iridoid monoterpene indole alkaloids which have previously been shown to act on nicotinic acetylcholine receptors, potassium channels, and calcium channels. Here, we isolated a new alkaloid from maqui leaves, now called makomakinol, together with the known alkaloids aristoteline, hobartine, and 3-formylindole. Moreover, the polyphenols quercetine, ethyl caffeate, and the terpenes, dihydro-β-ionone and terpin hydrate, were also obtained. In light of the reported analgesic and anti-nociceptive properties of *A. chilensis*, in particular a crude mixture of alkaloids containing aristoteline and hobartinol (PMID 21585384), we therefore evaluated the activity of aristoteline and hobartine on Na_V_1.8, a key Na_V_ isoform involved in nociception, using automated whole-cell patch-clamp electrophysiology. Aristoteline and hobartine both inhibited Nav1.8 with an IC_50_ of 68 ± 3 µM and 54 ± 1 µM, respectively. Hobartine caused a hyperpolarizing shift of the voltage-dependence of the activation, whereas aristoteline did not change the voltage-dependence of the activation or inactivation. The inhibitory activity of these alkaloids on Na_V_ channels may contribute to the reported analgesic properties of *Aristotelia chilensis* used by the Mapuche people.

## 1. Introduction 

Chronic pain is a complex phenomenon involving nociceptive, inflammatory, and neuropathic components. Voltage-gated sodium (Na_V_) channels are transmembrane proteins critical for the generation and propagation of action potentials in electrically excitable cells. Dysregulation of the Na_V_ channel expression and function in peripheral sensory neurons, which results in aberrant firing, contributes to the pathology of chronic pain [1,2,3]. The Na_V_1.8 subtype is abundantly expressed in nociceptive (pain-sensing) neurons and, as such, is an attractive target for the treatment of pain [4,5,6,7].

*Aristotelia chilensis* is a tree native to Chile commonly called “maqui”. The Mapuche people in Chile use the leaves and fruits of *Aristotelia chilensis* as an anti-inflammatory for the treatment of digestive ailments, fever, and skin lesions [8]. Maqui fruits are black berries containing a high concentration of polyphenols that are promoted to have many potential health benefits, including antioxidant, cardioprotective, and anti-inflammatory properties [9]. The leaves of *Aristotelia chilensis* contain non-iridoid monoterpene indole alkaloids including aristoteline, aristotelone, aristone, aristoquinoline, hobartine, 8-oxohobartine, and 8-oxo-9-dehydromakomakine, some of which have inhibitory activity on the human nicotinic acetylcholine receptors (nAChR) subtypes α3β4, α4β2, and α7 [10,11]. 

Aristoteline is pharmacologically the most well-characterized maqui alkaloid, with potent activity at the nAChR subtype α3β4 (IC_50_ of 0.4 μM), which is a potential therapeutic target for the treatment of addiction and depression [12]. Aristoteline also causes vasodilation (IC_50_ 16 μM) via the activation of potassium channels and the inhibition of Ca_V_1.2 in rodent vascular smooth muscle ex vivo, although the activity at the potassium and Ca_V_ channels in heterologous expression systems remains to be assessed [13]. Since aristoteline has broad activity at a range of ion channels, we hypothesized the maqui alkaloids may also have activity at Na_V_ channels.

In this study, we isolated eight natural compounds from the leaves of *A. chilensis*, including a new alkaloid called makomakinol, together with aristoteline, hobartine, and 3-formylindole. Moreover, the polyphenols quercetin, ethyl caffeate, and the terpenes dihydro-β-ionone and terpin hydrate were also obtained for the first time from this plant. The effects of aristoteline and hobartine on Na_V_1.8 and Na_V_1.7 were evaluated using automated whole-cell patch-clamp electrophysiology. Here, we describe for the first time that aristoteline and hobartine inhibit Na_V_ channels with a micromolar potency.

## 2. Results 

### 2.1. Chemical Characterization of Purified Compounds from Aristotelia chilensis Leaves 

Purification of secondary metabolites from *Aristotelia chilensis* leaves yielded eight natural compounds (see the structures in Figure 1). Four indole alkaloids identified as 3-formylindole (24 mg, gummy oil, 0.0003% yield); aristoteline (600 mg, yellow crystals, 0.0075% yield); hobartine (100 mg, colorless crystals, 0.00125% yield); and a new indole alkaloid called makomakinol (2.1 mg, white powder, 0.000026% yield). Between the non-alkaloids isolated are the terpenes dihydro-β-ionone (50 mg, yellow gummy, 0.0006% yield) and terpin hydrate (300 mg, colorless crystals, 0.0036% yield), together with the polyphenols ethyl caffeate (10 mg, colorless crystals, 0.00012% yield) and quercetin (100 mg, white solid, 0.0012% yield). 

Makomakinol has a molecular formula of C_20_H_22_N_2_O_2_, *m*/*z* 322.1 and appears as a white powder (MeOH); [α]_D_^25^: +68.76 (*c*: 0.05, MeOH). Its structure was determined via 1D and 2D NMR analysis, as shown in Table 1. 

The NMR spectra of makomakinol are very similar to those of hobartine, with the following differences: (i) a carbonyl group instead of a CH_2_-group at position 8; (ii) a quaternary sp^2^ carbon instead of an sp^3^-CH at position 9; (iii) an exocyclic sp^2^-CH_2_-group connected to a quaternary sp^2^-C instead of a methyl group at positions 17 and 11, resp.; and (iv) an additional OH group which could be located at position 16.

The absolute configuration of makomakinol was determined via circular dichroism (CD) and an analysis of coupling constants. First, the CD spectra of several possible stereoisomers of makomakinol were calculated in silico and compared to the experimental CD spectrum measured for naturally occurring makomakinol, as shown in Figure 2A. Only the CD spectra calculated for the (12*S*,14*R*,16*R*)- and for the (12*S*,14*R*,16*S*)-isomer of makomakinol are in good agreement with the experimental CD-spectrum, as shown in Figure 2B. To distinguish between these two stereoisomers, the coupling constants in the ^1^H-NMR spectrum of natural makomakinol were analyzed, as shown in Figure 2C. In natural makomakinol, the coupling constants of 11.9 Hz and 5.1 Hz from H-16 to both of the H-15 protons indicate a *trans* (11.9 Hz, dihedral angle of 168.3°) and a *gauche* (5.1 Hz, dihedral angle of 53.6°) configuration between these pairs of protons, which is only possible with H-16 in an axial position (16*S*-configuration). For the (12*S*,14*R*,16*R*)-isomer, which would also be in agreement with the experimental CD-spectrum, dihedral angles of 39.3° and 75.0° between H-16 and the two protons H-15 were determined from the calculated minimum structure, as shown in Figure 2D. Both dihedral angles are not in line with the observed coupling constants. From these considerations, it can be concluded that the absolute configuration of natural makomakinol is 12*S*,14*R*,16*S*.

### 2.2. Hobartine and Aristoteline Inhibit Na_V_ Channels 

The Hobartine and aristoteline concentration dependently inhibited the peak current of Na_V_1.8 with an IC_50_ of 54 ± 1 µM and 68 ± 3 µM, respectively (Figure 3A,D). To assess if this inhibition was state-dependent, we used an 8 s conditioning voltage step with −40 mV to inactivate approximately half of the available channels. Using this protocol, the potency of hobartine (IC_50_ 20 ± 2 µM) and aristoteline (IC_50_ 30 ± 6 µM) only slightly increased, indicating that unlike local anesthetics, they show a minimal preference for the inactivated state (Figure 3B,D). To assess if hobartine and aristoteline are selective for the Na_V_1.8 isoform, we also tested the activity at Na_V_1.7. Both hobartine (IC_50_ 69 ± 3 µM) and aristoteline (49 ± 7 µM) inhibited the Na_V_1.7 peak current with a similar potency to Na_V_1.8 (Figure 3C,F).

### 2.3. Mechanism of Hobartine and Aristoteline Na_V_1.8 Block 

To gain insights into the mechanism of the Na_V_ channel block, we next assessed the effect of hobartine and aristoteline on the voltage–current relationship of Na_V_1.8 (Figure 4A,D). At a concentration of 50 µM, hobartine caused a hyperpolarizing shift of the voltage-dependence of the activation (V_1/2_: control 0.4 ± 1.3 mV, hobartine −8.6 ± 0.8 mV; *p* < 0.05 paired *t*-test; Figure 3B) but had no significant effect on the voltage-dependence of the steady-state fast inactivation (V_1/2_: control −36.4 ± 3.0 mV, hobartine −41.1 ± 3.8 mV; *p* > 0.05 paired *t*-test; Figure 3C). In contrast, aristoteline at 50 µM had no effect on the voltage-dependence of the activation (V_1/2_: control 1.0 ± 0.0 mV, aristoteline 1.0 ± 0.0 mV; *p* < 0.05 paired *t*-test; Figure 4E) or the voltage-dependence of the steady-state fast inactivation (V_1/2_: control −33.3 ± 2.3 mV, aristoteline −34.9 ± 1.4 mV; *p* > 0.05 paired *t*-test; Figure 4F). 

## 3. Discussion

*Aristotelia chilensis* has been widely used by the Mapuche people due its anti-inflammatory and analgesic properties. The topical administration of crude leaf extract to the tails of mice has previously been demonstrated to be anti-nociceptive in response to radiant heat and formalin [14,15]. Therefore, in this study, we purified the main chemical constituents of a leaf extract of *Aristotelia chilensis*, finding a new alkaloid, makomakinol, as part of the minor constituents of the plant with a 0.000026% yield. The absolute configuration (12*S*,14*R*,16*S*) was proposed by a combination of the coupling constant analysis and the CD spectra. Moreover, this is the first time that dihydro-β-ionone and terpin hydrate were isolated from this plant. Dihydro-β-ionone or 2,4,4-trimethyl-3-(3-oxobutyl)cyclohex-2-en-1-one (CAS number 72008-46-9) is a small molecule used in perfumery with an odor description of floral, violet, pine, and woody [16]. Naturally, dihydro- β-ionone is part of the volatile constituents of plants, which plays a defense role against herbivores; for example, in canola (*Brassica napus*), this compound is released by the plant after herbivores attack, acting as a repellent [17]. On the other hand, terpin hydrate was used as an expectorant for bronchitis, but its use is controversial after the FDA found a lack of evidence for its effectiveness [18]. Together with the phytochemical characterization of the plant, we evaluated the effect of aristoteline and hobartine on Na_V_1.7 and Na_V_1.8, which are Na_V_ subtypes expressed in peripheral sensory neurons and important analgesic targets.

Consistent with the reported analgesic activity of *Aristotelia chilensis,* both aristoteline and hobartine inhibited Na_V_1.8 with a micromolar potency. In addition, both indole alkaloids inhibited Na_V_1.7 with a similar potency, indicating that they are unlikely to exhibit selectivity for any Na_V_ subtype, although this remains to be assessed. The effects of hobartine on the electrophysiological properties of Na_V_1.8 appear to be distinct from those of simple Na_V_ pore blockers as it exhibited an effect on the voltage-dependence of the activation, shifting the V_1/2_ to more negative potentials (Figure 4B) that are also distinct from those of local anesthetics which typically exhibit a more pronounced preference for the inactivated state (Figure 3B). These results agree with those obtained from a representative of the β-carboline alkaloids found in *P. quassiodes*, DHCT, as it also robustly shifted the voltage-dependent activation to hyperpolarized potentials [19,20,21]. In contrast, the small molecule local anaesthetic lidocaine produces a hyperpolarizing shift in voltage-dependent inactivation [22]. On the other hand, aristoteline blocked the Na_V_1.8 current without affecting the voltage-dependence of the activation or inactivation (Figure 4E,F), suggesting that the inhibition may result from a physical blockage of the sodium ion pathway (pore blocker) in a manner similar to tetrodotoxin (TTX) and saxitoxin (STX), two marine alkaloids known to inhibit Na_V_ channels [23,24]. The activity of aristoteline and hobartine at Na_V_1.8 and 1.7, two channel subtypes expressed in peripheral nociceptors, may contribute to the reported analgesic activity of *Aristotelia chilensis* leaf extracts.

## 4. Material and Methods

### 4.1. General Information

Column chromatography was performed using Merck silica gel 60 and Sephadex LH-20 (25−100 μm; Aldrich, Santiago, Chile). The progress of purification was followed by using analytical thin-layer chromatography (TLC) from Merck Silica Gel 60F254 sheets (Darmstadt, Germany) together with Low Field NMR (LF-NMR, Bruker 80 Benchtop, Germany). TLC were eluted with a mixture of solvents as n-hexane (hex), ethyl acetate (EtOAc), and methanol; evaluated using UV light (254 nm); then stained with Dragendorff and/or with KMnO_4_. Solvents and fractions were concentrated in a rotavap Büchi R100 at 45 °C. Solvents used in this study were distilled prior to use and dried over appropriate drying agents. 

### 4.2. Purification of Secondary Metabolites from Leaves of Aristotelia chilensis

A total of 8 kg of leaves of *Aristotelia chilensis* (maqui) were collected in Temuco, beginning the summer season in December 2018. Vegetal material was powdered and macerated for 3 days in water acidified with HCl to pH 3 at room temperature. The aqueous–acid layer was extracted with EtOAc and the organic solvent was evaporated under reduced pressure at 45 °C and 200 mbar to afford a gummy residue (total acid extract 110 g). Then, the acid water was alkalinized to pH 11 with NaHCO_3_-NaOH and subsequently extracted with EtOAc, 4 L, 3-fold. After solvent evaporation, 50 g of a gummy red extract rich in alkaloids were obtained. The crude alkaloid extract was chromatographed using a silica gel column (200–300 mesh) and an increased solvent polarity, from hex 100% to EtOAc 100% to isopropanol 100%, giving fractions F1 to F6. Fraction F1 (17 g) only contained fatty acids. F2 (10 g) was eluted with hex:EtOAc 3:2 *v*/*v* giving 3-formylindole (24 mg, gummy oil, 0.0003% yield). The fraction F3 (12 g) was further purified via silica gel chromatography using hex:EtOAc 1:1 *v*/*v*, giving aristoteline, (600 mg, yellow crystals, 0.0075% yield). The fraction F4 (5 g) was applied to a Sephadex LH-20 column and eluted with isopropanol, giving hobartine (100 mg, colorless crystals, 0.00125% yield). The fraction F6 (2 g) gave makomakinol (2.1 mg, white powder, 0.000026% yield). 

The total acid extract (110 g) was purified via a silica gel column using hex:EtOAc 2:1 *v*/*v*, giving the terpenes dihydro-β-ionone (50 mg, yellow gummy, 0.0006% yield), followed by terpin hydrate (300 mg, colorless crystals, 0.0036% yield). Then, using hex:EtOAc 1:1, two polyphenols were identified as ethyl caffeate (10 mg, colorless crystals, 0.00012% yield) and quercetine (100 mg, white solid, 0.0012% yield).

### 4.3. Identification of Compounds Purified from Aristotelia chilensis

All compounds were analyzed via GC-MS (Schimadzu GC-QP2020NX, Kyoto, Japan) and their fragmentation mass spectra were compared with the library NIST2017. The compounds 3-formylindole, dihydro-β-ionone, and terpin hydrate gave a correlation of over a 90% similarity to the NIST library. All compounds were determined via 1D and 2D nuclear magnetic resonance (NMR). The NMR data of aristoteline and hobartine were compared with the pure standard available in the laboratory of the author (C.P.) with excellent agreement to the previously reported NMR data [12]. 

The new alkaloid makomakinol was fully evaluated via 1D and 2D NMR. The ^1^H- and ^13^C NMR spectra were recorded in a CD_3_OD solution in 5 mm tubes at RT on a Bruker Avance III 600 MHz spectrometer (Bruker Biospin GmbH, Rheinstetten, Germany) at 600.13 (^1^H) and 150.61 (^13^C) MHz, with the deuterium signal of the solvent as the lock and TMS (for ^1^H) or the solvent (for ^13^C) as the internal standard. All spectra (^1^H, ^13^C, gs-H,H−COSY, edited HSQC, gs-HMBC, and NOESY) were acquired and processed with the standard Bruker software Topspin 4.3.0. Optical rotations were recorded on a Dichrom Model P-2000 polarimeter. The CD spectra were observed with an JASCO J-815 spectralpolarimeter.

### 4.4. Theoretical Calculations

Different conformations and configurations of the compound were optimized at the MP2/6-311G** level of theory without any restrictions. The ECD were computed using the Time Dependent DFT (TDDFT) [25,26] algorithm in the program package GAUSSIAN 09 [27]. The B3LYP functional and 6-31G* basis set was applied [28]. A total of 10 singlet and 10 triplet states were solved (keyword TD (NStates = 10, 50–50). All GAUSSIAN 09 results were analyzed and the spectra were displayed using the SpecDis 1.62. The molecule is displayed using SYBYL-X 2.1.1 (2013).

### 4.5. Cell Culture

Human embryonic kidney (HEK) 293 cells stably expressing human Na_V_1.7/β1 (SB Drug Discovery, Glasgow, UK) or Chinese hamster ovary (CHO) cells stably expressing human Na_V_1.8/β3 in a tetracycline-inducible system were cultured in a minimum essential media (MEM) supplemented with 10% fetal bovine serum (FBS), 2 mM l-glutamine, and selection antibiotics as recommended by the manufacturer. To induce hNa_V_1.8 expression, tetracycline (1 μg/mL) was added to the culture media 72 h prior to the functional assays. The cells were passaged every 3–4 days after reaching a 70–80% confluence using TrypLE Express (Thermo Fisher Scientific, Scoresby, VIC, Australia) and grown in an incubator at 37 °C with 5% CO_2_. Prior to the electrophysiology experiments, the cells were dissociated with TrypLE Express and resuspended in Dulbecco’s Modified Eagle Medium (DMEM) with 25 mM HEPES, 100 U/mL penicillin-streptomycin, and 0.04 mg/mL of a trypsin inhibitor from Glycine max (soybean), then stirred for 30 min. 

### 4.6. Electrophysiology

Whole-cell patch-clamp experiments were performed on a QPatch-16 automated electrophysiology platform (Sophion Bioscience, Ballerup, Denmark) using single hole (QPlate 16 with a standard resistance of 2 ± 0.4 MΩ for Na_V_1.7) or multi-hole (QPlate 16× with a standard resistance 0.2 ± 0.04 MΩ for Na_V_1.8). Whole-cell currents were filtered at 8 kHz and acquired at 25 kHz and the linear leak was corrected using P/4 subtraction (leak potential −90 mV, leak sweep amplitude 10%). 

The extracellular solution contained in mM included NaCl 140 (NaCl 70/Choline Chloride 70 for Na_V_1.7), KCl 4, CaCl_2_ 2, MgCl_2_ 1, HEPES 10, and glucose 10; pH 7.4; osmolarity 305 mOsm. TTX (1 μM) was added to the extracellular solution for the Na_V_1.8 recordings to inhibit endogenous TTX-sensitive Na_V_ currents in CHO cells. The intracellular solution contained in mM included CsF 140, EGTA/CsOH 1/5, HEPES 10, and NaCl 10; pH 7.3 with CsOH; osmolarity 320 mOsm. Compounds were made up to a 100 mM stock in DMSO and diluted in an extracellular solution at the concentrations stated. Concentration–response curves at the resting state were acquired using a holding potential of −90 mV and a 50 ms test pulse to −20 mV (for Na_V_1.7) or +10 mV (for Na_V_1.8) every 20 s (0.05 Hz). Concentration–response curves at the inactivated state were acquired using a holding potential of −90 mV, an 8 s conditioning pulse to −40 mV, a 20 ms recovery pulse to −90 mV, and a 50 ms test pulse to +10 mV every 20 s (0.05 Hz). Peak current was normalized to the buffer control after 2 min incubations of increasing concentrations of the compound (1–100 μM). I–V curves were obtained with a holding potential of −90 mV and a series of 500 ms step pulses that ranged from −80 to +90 mV in 5 mV increments, followed by a 20 ms test pulse to +10 mV to assess the voltage-dependence of the steady-state fast inactivation (repetition interval 5 s). Peak currents were normalized to the buffer control before and after the 5 min incubation of the compound (50 μM).

### 4.7. Data Analysis

The data were plotted and analyzed via GraphPad Prism, version 9.5.1. Statistical significance was defined as *p* < 0.05 and was determined via the paired *t*-test. Concentration–response curves were fitted with a four-parameter Hill equation with the variable Hill coefficient. Conductance–voltage curves were generated by calculating the conductance (G) at each voltage (V) using the equation G = I/(V − V_rev_), where V_rev_ is the reversal potential. Conductance–voltage and steady-state fast inactivation relationships were fitted using a Boltzmann equation. The data are expressed as the mean ± standard error of the mean (SEM).

This information contributes to understanding the role of natural compounds in the traditional medicine of the pre-Columbian people and how these indoyl alkaloids can inhibit pain via the inhibition of Na_V_ channels, which could be useful for the design of new ligands based on this structure. 

## Figures and Tables

**Figure 1 ijms-24-15504-f001:**
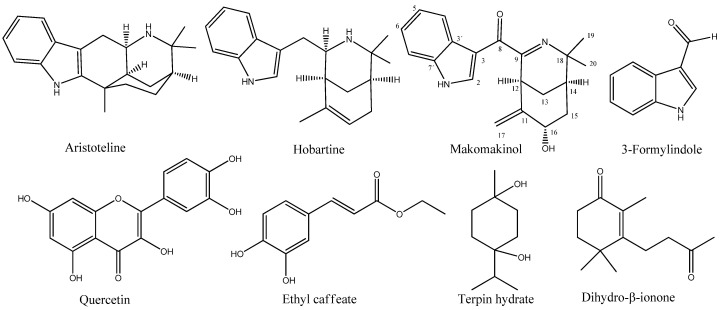
Chemical structure of natural compounds isolated from *Aristotelia chilensis* leaves, including the alkaloids aristoteline, hobartine, 3-formylindole, and the numbered structure of the new alkaloid makomakinol, together with the polyphenols quercetin and ethyl caffeate, and the terpenes terpin hydrate and dihydro-β-ionone.

**Figure 2 ijms-24-15504-f002:**
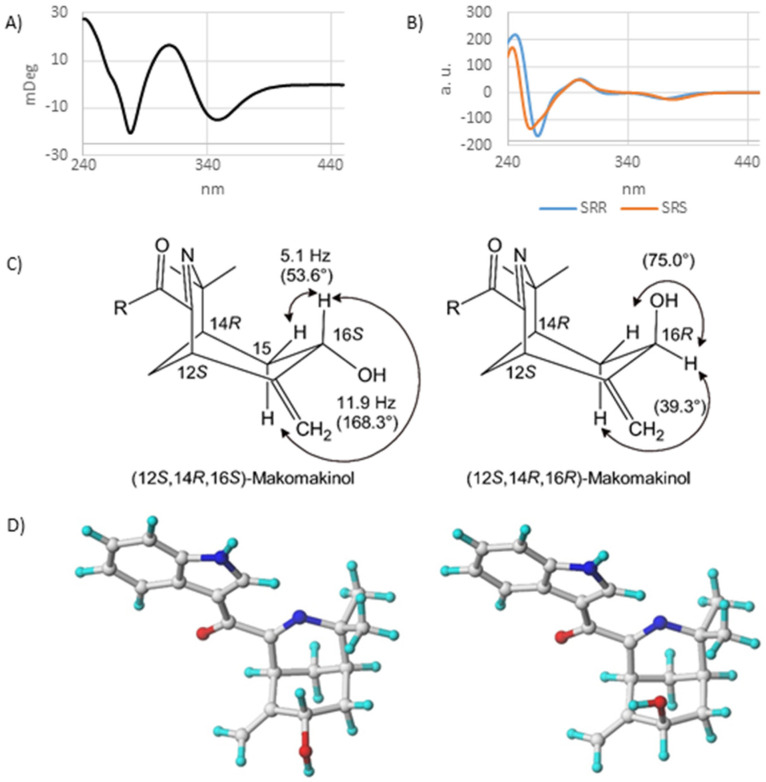
(**A**) Experimental CD spectrum for natural makomakinol. (**B**) Calculated CD spectra for (12*S*,14*R*,16*S*)– and (12*S*,14*R*,16*R*)–diastereomers of makomakinol. (**C**) Structures with coupling constants and calculated dihedral angles (in parentheses) of (12*S*,14*R*,16*S*)– and (12*S*,14*R*,16*R*)–diastereomers of makomakinol. (**D**) Calculate minimum structures of (12*S*,14*R*,16*S*)– (**left**) and (12*S*,14*R*,16*R*)–diastereomers (**right**) of makomakinol (ball–stick model, nitrogen in blue and oxygen in red color).

**Figure 3 ijms-24-15504-f003:**
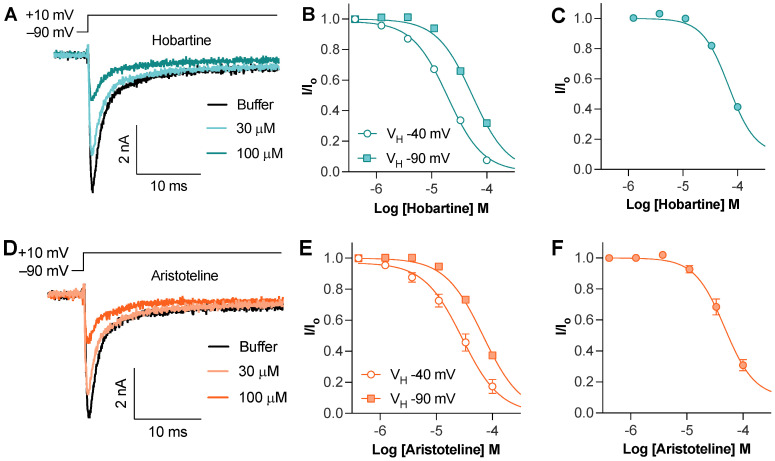
Effect of maqui alkaloids on voltage–gated sodium channels assessed via whole–cell patch–clamp electrophysiology. (**A**) Representative hNa_V_1.8 trace before and after addition of hobartine. Currents were elicited via a 50 ms voltage step to 10 mV from −90 mV holding potential. (**B**) Horbartine concentration dependently inhibited Na_V_1.8 peak current elicited via a 50 ms voltage step to 10 mV from −90 mV holding potential (IC_50_ of 54 ± 1 µM; *n* = 5) or after an 8 s condition voltage step to −40 mV to inactivate approximately half of the available channels (IC_50_ of 20 ± 2 µM; *n* = 5). (**C**) Hobartine concentration dependently inhibited Na_V_1.7 peak current elicited via a 50 ms voltage step to −20 mV from −90 mV holding potential (IC_50_ of 69 ± 3 µM; *n* = 4). (**D**) Representative hNa_V_1.8 trace before and after addition of aristoteline. Currents were elicited via a 50 ms voltage step to 10 mV from −90 mV holding potential. (**E**) Aristoteline concentration dependently inhibited Na_V_1.8 peak current elicited via a 50 ms voltage step to 10 mV from −90 mV holding potential (IC_50_ of 68 ± 3 µM; *n* = 5) or after an 8 s condition voltage step to −40 mV to inactivate approximately half of the available channels (IC_50_ of 30 ± 6 µM; *n* = 3). (**F**) Aristoteline concentration dependently inhibited Na_V_1.7 peak current elicited via a 50 ms voltage step to −20 mV from −90 mV holding potential (IC_50_ of 49 ± 7 µM; *n* = 5).

**Figure 4 ijms-24-15504-f004:**
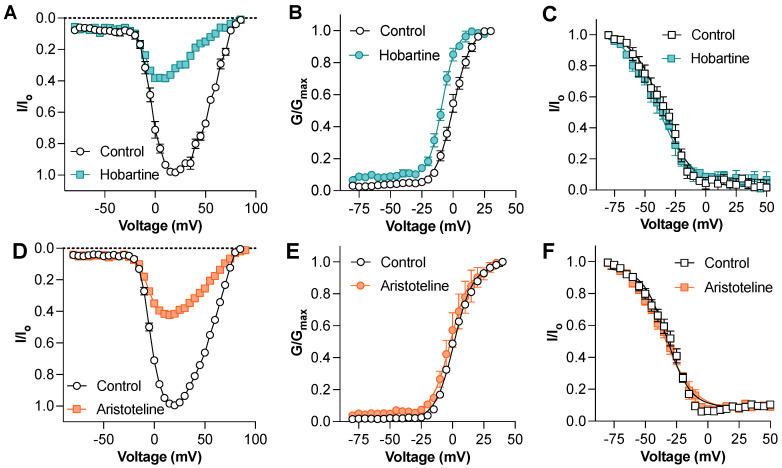
Mechanism of action of hobartine and aristoteline inhibition of Na_V_1.8 currents (**A**) Na_V_1.8 current–voltage relationship before and after addition of 50 µM hobartine (*n* = 5). (**B**) Na_V_1.8 conductance–voltage relationship before and after addition of 50 µM hobartine (*n* = 5). Hobartine shifted the V_1/2_ of voltage-dependence of activation by −9.0 mV. (**C**) Na_V_1.8 voltage-dependence of steady-state fast inactivation before and after addition of 50 µM hobartine (*n* = 4). Effect of aristoteline on voltage-gated sodium channels assessed via whole-cell patch-clamp electrophysiology. (**D**) Na_V_1.8 current–voltage relationship before and after addition of 50 µM aristoteline (*n* = 5). (**E**) Na_V_1.8 conductance–voltage relationship before and after addition of 50 µM aristoteline (*n* = 5). (**F**) Na_V_1.8 voltage-dependence of steady-state fast inactivation before and after addition of 50 µM aristoteline (*n* = 5).

**Table 1 ijms-24-15504-t001:** NMR data of makomakinol (^1^H-NMR at 600 MHz in MeOD, δ in ppm, *J* in Hz; ^13^C-NMR at 150 MHz in MeOD).

Carbon	^13^C (ppm)	^1^H (ppm)	HMBC
	138.3	8.09, *s*	3, 8, 3′, 7′
3	115.7	-	
3′	127.7	-	
4	123.0	8.28, *d*, *J* = 7.5 Hz	3(w), 6, 7(w), 3′(w), 7′
5	123.6	7.24, *m*	
6	124.7	7.24, *m*	
7	113.0	7.46, *d*, *J* = 7.4 Hz	3′, 7′(w)
7′	138.3	-	
8	189.8	-	
9	169.0	-	
11	149.8	-	
12	42.0	3.90, *br*	9, 11, 14, 16, 17
13	30.3	2.21, *ddd*, *J* = −12.6, 6.2, 2.8 Hz 1.75, *ddd*, *J* = −12.6, 3.3, 2.3 Hz	12, 15 9, 12, 14, 18
14	38.6	2.08, *m*	12, 13, 15, 16, 18, 19
15	40.5	1.52, *m* 2.41, *ddt*, *J* = −13.0, 5.6, 2.8 Hz	11, 13, 14, 16, 18, 19 11, 13, 14, 16
16	68.3	4.21, *ddt*, *J* = 11.9, 5.1, 2.5 Hz	11(w), 12, 16
17	108.4	5.00, *br s* 4.92, *br s*	11(w), 12, 16 11(w), 12, 16
18	60.3	-	
19	27.0	1.53, *s*	14, 18, 20
20	31.0	1.33, *s*	14, 18, 19

## Data Availability

Not applicable.

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
