# Peer review of "Natural Compounds Purified from the Leaves of Aristotelia chilensis: Makomakinol, a New Alkaloid and the Effect of Aristoteline and Hobartine on NaV Channels"

_ijms, 2023, doi:10.3390/ijms242115504_

Round 1
Reviewer 1 Report
The manuscript ijms-2653176 by Pérez et al., entails the phytochemical characterization of Aristotelia chilensis including the isolation of a new alkaloid called makomakinol. Moreover, the effect of two of its alkaloids i.e. aristoteline and hobartine was evaluated on voltage-gated sodium channels Nav1.8 and Nav1.7. The manuscript is well written, and the results are clearly and thoroughly presented.
1. Introduction (last paragraph): were also obtained for first time from this plant = were also obtained for the first time from this plant.
2. Could you improve the analysis of Figure 1?
3. Discussion: due its medicinal properties as an anti-inflammatory and analgesic = due to its anti-inflammatory and analgesic properties.
4. I believe references 8, 32 and 33 should be corrected and written according to the journal’s guidelines.
Author Response
Dear reviewer,
I would like to thank your comments and suggestions, which were considered in our corrected manuscript.
Sincerely,
Dr. Cristian Paz
Here we give you a response about your observations:
- Introduction (last paragraph): were also obtained for the first time from this plant = were also obtained for the first time from this plant.
Response: The sentence was corrected
- Could you improve the analysis of Figure 1?
Response: We improved this part
- Discussion: due to its medicinal properties as an anti-inflammatory and analgesic = due to its anti-inflammatory and analgesic properties.
Response: the change was done
- I believe references 8, 32, and 33 should be corrected and written according to the journal’s guidelines.
Response: All the references were checked and corrected.
Reviewer 2 Report
Revision Report
The manuscript entitled “Natural compounds purified from leaves of Aristotelia chilensis: makomakinol a new alkaloid and effect of aristoteline and hobartine on NaV channels” is good written. However, it requires minor revision before acceptance.
There are some typographical mistakes. These should be corrected.
In section 4.2, the line 10 has extra word “no”.
There should be the weight of the fractions from which compounds are isolated.
In section 4.3, what means CP Library?
These identified compounds are mostly polar. How these are identified with GC-MS?
There is no concluding paragraph at the end of manuscript.
“Møller, P. MP2 Notes. Eur. J. Cardio-thoracic Surg. 1934, 53, 1237–1243” This reference is very old. It should be replaced with newer one.
Few mistakes
Author Response
Dear Reviewer,
I would like to thank your comments and suggestions, which were considered in our corrected manuscript.
warm regards,
Dr. Cristian Paz
- There are some typographical mistakes. These should be corrected.
Response: we found minor misspellings which were corrected.
- In section 4.2, the line 10 has extra word “no”.
Response: the sentence was corrected
- There should be the weight of the fractions from which compounds are isolated.
Response: The weight was added to each fraction.
- In section 4.3, what means CP Library?
Response: The NMR data of aristoteline and hobartine were compared with pure standard available in the laboratory of the author (C.P.)
- These identified compounds are mostly polar. How these are identified with GC-MS?
Response: yes, the compounds are polar for the silica gel, but we have to consider that they are formed by 20 Carbons and only 2 nitrogens. They have a good detection by CG-MS, and also was possible to see the molecular ion by this analytical technique.
- There is no concluding paragraph at the end of the manuscript.
Response: A concluding paragraph was added before “author contribution”.
- “Møller, P. MP2 Notes. Eur. J. Cardio-thoracic Surg. 1934, 53, 1237–1243” This reference is very old. It should be replaced with a newer one.
Response: This reference was deleted.